# OpenReview forum: "Hail to the Thief: Exploring Attacks and Defenses in Decentralized GRPO"
_ICLR.cc/2026/Conference — ICLR 2026 Conference Withdrawn Submission_

### Official Review · Reviewer_vDVt · 2025-10-24

**Soundness:** 3
**Presentation:** 3
**Contribution:** 2
**Rating:** 4
**Confidence:** 3

**Summary:**

The paper introduces the first adversarial attacks and defenses for decentralized GRPO. It shows that malicious nodes in decentralized GRPO can poison others through sharing adversarial completions either by inserting unrelated text (out-of-context) or manipulating task logic or code (in-context). Experiments on math and coding datasets demonstrate attack success rates up to 100% within a few iterations. The authors also propose two defenses which are shown to be effective.

**Strengths:**

- I find the idea of the paper novel, underexplored and interesting. It insightfully highlights how reward models that focus only on final answers can overlook flawed reasoning steps, leading to incorrect reward assignments and reinforcing undesirable reasoning behaviors. I find this both intuitive and exciting to think about.

- The paper’s writing logic and presentation are clear, well-organized, and engaging, providing readers with a smooth and enjoyable experience.

- All the claims and ideas are beautifully implemented and supported by comprehensive experiments, which I find both convincing and valuable.

**Weaknesses:**

- Despite the interesting results, I have serious concerns about the methodology and underlying assumptions. The attacker nodes are assumed to have access to both oracle responses and the reward function, enabling them to easily craft poisoned completions, a setting that feels unrealistic to me. Moreover, some experiments are heavily dependent on specific datasets and prompt availability, as even the authors note that *“in horizontal RL is that the occurrence of the targeted equation is rare in the entire GSM8k dataset. As this specific attack is very dependent on the prompt it might not succeed well when the attacker cannot pick and choose their questions”*.
All these assumptions and access levels make the results somewhat expected rather than surprising. A more realistic scenario would avoid giving attackers oracle or reward access and instead explore subtler injection methods, such as prompt-level manipulations that preserve final answers to maintain high rewards, though implementing such attacks would be significantly more challenging and warrant deeper investigation.

- The paper would benefit from additional proofreading. For instance, in Line 245, the sentence “In an out-of-context attack, we aim to inject arbitrary malicious text. and the same text.” contains a punctuation error. Additionally, Footnote 6, which reads “But it’s not, maybe not,” appears to be accidental or misplaced and I'm not sure if I understand it.

- Overall, I think a deeper analysis of the effectiveness and trade-offs from the computational side of the defenses, and how scalable and generalizable they are, would have been helpful. For instance, the LLM-as-a-Judge defense is computationally expensive, as it requires evaluating every generated completion individually with a separate model, significantly increasing inference cost and latency during decentralized training.

**Questions:**

Please refer to the weaknesses.

---

> ### Author Response · Authors · 2025-11-14
> **Response to Reviewer vDVt for the paper "Hail to the Thief"**
>
> We would like to thank the reviewer for their comments and issues raised, as they help us clarify the ideas in our paper.
>
> We are currently running the following experiments to provide additional experimental results, and will share the results as soon as possible:
> - **(Attack) with less malicious participation:** '2 + 2 = 5' attack with 12.5% malicious participation
> - **(Attack) against larger model:** '2 + 2 = 5' attack on QWEN 3B model.
> - **(Defense) LLL-as-a-judge without additional judge model:** Using the trained model to judge completions instead of an auxiliary one.
>
> Below we address the issues raised by the reviewer.
>
>
> **W1** *On the assumptions of the paper*
>
> Here, we motivate our assumptions and choice of setting. In the final version of the paper, we will restructure the motivation to better emphasize our points. Our setting is modeled after real-world applications such as RL Swarm, where nodes exchange completions and locally reward them to use in a GRPO-style training. As we focus on tasks where verifiable rewards are used, nodes must have access to at least the correct final answer, which they can use as ground truth comparison when rewarding completions. This is the bare minimum required for such RL training to work. Thus, an attacker would also have access to the correct final answer (as would all nodes). The only way to avoid this is to defer rewarding to a delegated authority, which would be the only party that has the correct answers. However, this moves away from decentralized solutions where nodes can operate independently and instead introduces a new potential attack where the attacker is said delegated authority. This being said, access to "correct" completions is not a must-have assumption - it merely makes the attack easier. As stated in Section 2, an attacker could alternatively use an already poisoned model to produce completions that contain the given attack (for example having a model already trained to produce 2 + 2 = 5) or can pick a list of questions and manually construct a dataset of poisoned completions. Both of these are feasible in a real-world scenario.
>
>
> **W2** *On spelling mistakes*
>
> We thank the reviewer for pointing out the spelling mistakes. We will address the spelling mistakes mentioned. We wish to clarify that the footnote mentioned is merely a reference to the song 2+2=5 from the album Hail to the Thief by Radiohead. As the attack examples were heavily inspired by this song, the footnote specifically refers to the last lines of the song "Oh, go and tell the king that the sky is falling in/ But it's not, maybe not". We can remove the footnote if reviewers find it confusing.
>
>
> **W3** *On the computational cost of the defenses*
>
> We thank the reviewer for the discussion on this point. We will amend our discussion to include the following analysis. The computational cost of homogeneous defense via checking token generation is a single forward pass per completion for auto-regressive models since causal masking allows to check all the generated tokens at once. Whereas, the cost of heterogeneous defense via LLM-as-a-Judge depends on the judge LLM. In our experiments with the LLM-as-a-Judge defense, we observe that training time is almost tripled (thus merely the judging aspect constitutes 66% of the actual training). As future work, we aim to improve the efficiency of this defense, by for example, combining all the completions into one prompt and asking the model to evaluate them together.

---

> ### Author Response · Authors · 2025-11-19
> **Further experimental results**
>
> # Additional Experiments
>
> Below, we list the additional experiments that we run based on the received reviews. Additional results show that (i) poisoning attacks are successful with less malicious participation, (ii) poisoning attacks are successful for larger models, and (iii) LLM-as-a-judge defense where each trained model itself is used to judge completions (instead of an auxiliary one) fails, as mentioned in the paper.
>
> All additional experiments support the claims made in the paper. We hope that these experiments address the comments of the reviewers, and we will reflect the results in the updated version of the paper. We are happy to answer any further comments the reviewer might have.
>
> ## 1. Attack with less malicious participation
>
> - **Goal:** Testing the effectiveness of the attack with less malicious participation.
> - **Setup:** We implement the '2 + 2 = 5' attack with 12.5% malicious participation on a Qwen-2.5 1.5B model. All other (hyper)parameters are kept identical.
>
> ### Results
>
> - **TL;DR:** Even with 12.5% malicious participation, the attack succeeds, albeit with a noticeably lessened ASR relative to 25% malicious participation.
> - **Figure:** The results are shown in [https://anonymous.4open.science/r/HTTF-1066/fig_less.png](https://anonymous.4open.science/r/HTTF-1066/fig_less.png)
>
>
> We theorized a similar effect in Appendix B.2, where the effect of the attack was nearly maximal at 50% malicious participation (as shown in Figure 8). In Figure 9 of Appendix B.2 we observed that with participation at 8% in Horizontal "Hail to the Thief" (equivalent to 1 poisoned completion per group), the attack did not succeed. This suggests that the ideal participation ratio can be extrapolated from Figure 8 to be at 50% malicious participation, though even at around 20% being sufficient.
>
> ## 2. Attack against larger model
> - **Goal:** Testing the effectiveness of the attack against larger models.
> - **Setup:** We implement the "2+2=5" attack (from Section 3.4.1) with the QWEN-2.5 3B model. All other (hyper)parameters are kept identical.
>
> ### Results
>
> - **TL;DR:** The attack works also for the larger models, but ASR is lower than smaller ones.
> - **Figure:** The results are shown in [https://anonymous.4open.science/r/HTTF-1066/fig_3b.png](https://anonymous.4open.science/r/HTTF-1066/fig_3b.png)
>
> As can be seen from the figure, the "2+2=5" attack is successful for the Qwen-2.5 3B model. Even after just 20 iterations, the ASR exceeds 50%. Moreover, the ASR of the 3B model is lower than that for the 1.5B model given in the paper. In Appendix B.3, we theorize that as models get better at the task, the attacker's success is diminished and thus requires a greater amount of poisoned completions.
>
>
>
> ## 3. Defense LLM-as-a-judge without additional judge model
> - **Goal:** Testing the effectiveness of the LLM-as-a-judge defense where the trained model itself is used to judge completions instead of an auxiliary one.
> - **Setup:** We implement the LLM-as-a-judge defense on the "2+2=5" attack by employing each trained model as its own judge model, rather than the auxiliary Llama model used in the paper. All other (hyper)parameters are kept identical.
>
> ### Results
> - **TL;DR:** The defense of using its own model for as a judge fails, as mentioned in the paper. The intuitive explanation of why the defense behaves poorly is that there is no feedback on which completions are malicious. Thus, the game theoretical optimal strategy for an agent is to accept every completion to maximize their rewards.
> - **Figure:** The results are shown in [https://anonymous.4open.science/r/HTTF-1066/fig_sd.png](https://anonymous.4open.science/r/HTTF-1066/fig_sd.png)
>
> The results demonstrate that over time the attack succeeds to almost 100% of the trained policy becoming poisoned. This defense fails for two main reasons. First, models initially struggled with the base task (solving math problems and adhering to some formatting), producing primarily incoherent gibberish. Thus, when judging incoming completions during the first few iterations, they seldom produced completions that contained a yes/no decision. Second, later on, models would not receive a reward signal on the yes/no decision completion they had produced. As such, across all of our experiments, the models collapsed to outputting only yes responses, accepting all malicious texts. Since it cannot determine which completions are malicious or benign, the policy never learns this behaviour well.

---

> ### Comment · Reviewer_vDVt · 2025-11-24
> **Response to Authors**
>
> Thank you for your detailed response.
>
> However, regarding the assumptions made in the paper, **I am still not fully convinced, and I noticed that Reviewer 2BAe raised similar concerns which make the attacks and the setup less realistic.** While I understand that GRPO training typically requires access to oracle answers and the reward model, this work is positioned as a security study, not a standard benign GRPO training setup. Under these assumptions, **the attacks and results feel much more expected than exciting**.
>
> In addition, I was unable to open any of the links you provided in your responses to other reviewers. Finally, **given that ICLR explicitly allows and encourages revisions to the PDF during the rebuttal period, I expected the authors to incorporate updates and new experiments directly into the paper rather than stating that changes will be made in the future.**
>
> For these reasons, I am maintaining my original score of 4.

---

> > ### Author Response · Authors · 2025-11-25
> > **Response to Reviewer vDVt**
> >
> > We thank the reviewer for their response. We have updated the current version of the PDF submission with changes highlighted in light blue for ease of access.
> >
> > All figures in our previous reply are included in the this revision with other additional modifications.

---

> ### Author Response · Authors · 2025-11-26
> **Reply to Reviewer vDVt**
>
> We have included in the revised manuscript a more detailed explanation on distributed RL for LLMs together with motivation for the threat model. We wish to motivate it here, in response to  the comment that "The attacker nodes are assumed to have access to both oracle responses and the reward function, enabling them to easily craft poisoned completions, a setting that feels unrealistic to me."
>
> We will do this in three parts. First, explaining distributed GRPO-style training. Second, we will motivate the attacker's capabilities. Lastly, we relate this to an already established field - federated learning.
>
> **Distributed GRPO-style training**
>
> As we study Reinforcement Learning with Verifiable Rewards (RLVR), nodes perform their training on some dataset, processing an input (question/prompt) and generating a completion. These completions are then shared across all nodes, together with the corresponding question. In line with decentralized systems, each model scores each incoming completion based upon some reward function. Typically, in RLVR GRPO training, these include validating whether the formatting is correct and whether the final answer (not the solution, just the answer) generated is the correct one. Thus, as a node must validate if the final answer is correct in a verifiable manner, it must have a ground truth label to compare it against. As current work has not explored utilizing different reward functions per different models, we assume the reward model is shared (i.e. everyone has it)
>
> **Attacker's capabilities**
>
> An attacker must have the same if not more capabilities than a benign node. As benign nodes require ground truth answer and a reward function for the training, otherwise no learning can occur, then an attacker must have access to these as well. An attacker does not require a ground truth solution - it just makes the attacker easier. Typical datasets used, such as GSM8, DeepMath-103k, etc. usually also include an example solution. Even if not present, an attacker can manually construct coherent solutions for the subset of the data, use an auxiliary model, or even make random reasoning up, as usually the reasoning is not checked by such sparse rewards.
>
> **Federated Learning**
>
> We relate homogeneous vertical dRL to Federated learning. Let's assume that for some FL task, nodes aim to learn a classifier on locally available data. For this, each node has some subset of the data $\mathcal{D}_i\subset\mathcal{D}$ containing an input and ground truth pairs $x,y$ and have a shared model $\theta$ and some loss function $\mathcal{L}$. In vertical homogeneous dRL, the loss function is equivalent to the reward mechanism, which guides the model in the correct direction, and the local dataset subset is the pairs of some question/prompt and some ground truth answer used in the reward function (equivalent to label). As we can see, if benign nodes (and thus attacker) do not have access to oracle answers (ground truth labels) and the reward model (the loss), even without an attack, the training cannot occur.
>
> We hope this motivates the setting and attacker's capabilities and answers the reviewer's final question and concern.

---

> > ### Comment · Reviewer_vDVt · 2025-11-27
> >
> > I appreciate the authors' fast reply. I know what Distributed GRPO training is and how it is carried out. The fact that your attacker nodes have the same access as other nodes makes the attack less realistic and less interesting. The results are expected because the model is receiving manipulated completions, which have been easily created by the attacker. Therefore, my opinion on the practicality and realism of the attacks remains the same. To avoid repetition, please check my previous responses.

---

> ### Author Response · Authors · 2025-11-28
> **On the Attack Scenario**
>
> We thank the reviewer for their response.
>
> In a distributed setting in which an adversary participates in a protocol (in this case, GRPO), it is standard to assume that the adversary possesses at least the capabilities required to execute the protocol, similar to all legitimate participants. It would be inconsistent to assume that an adversary engaged in the protocol lacks the fundamental knowledge necessary for participation - independent of any additional capabilities required to carry out an attack.
>
> An analogous situation in federated learning would be to assume that an adversarial participant is part of the FL protocol, yet restrict them from receiving the global model updates that all nodes are, by definition, intended to obtain.

---

### Official Review · Reviewer_AetF · 2025-10-29

**Soundness:** 2
**Presentation:** 3
**Contribution:** 3
**Rating:** 4
**Confidence:** 3

**Summary:**

The paper introduces a backdoor attack on decentralized GRPO, where some of the nodes generating rollouts are malicious. The key observation is that for outcome-based rewards (such as math answer correctness), the reasoning trace doesn't directly affect the reward, but it is still reinforced for high-reward samples. Thus an attacker can generate samples with malicious reasoning traces but correct final answers to inject undesired behaviors into the model.
The paper shows that this attack works to insert simple behaviors in math and code settings and also explores two defenses (one based on off-policy detection, one using an LLM to judge answer correctness).

**Strengths:**

Backdoor attacks on decentralized training are an interesting threat model. The key idea that the attacker can reinforce malicious intermediate steps in RL from outcome-based rewards is also interesting and new to me (though I'm not familiar enough with the field to be confident how novel it is).
The code injection attack in section 3.4.2 is the most compelling to me, since there is a relatively straightforward path from injecting malicious coding behavior to achieving various attacker goals. Overall, the ideas in this paper made me take these types of backdoor attacks more seriously as a threat foe decentralized RL.

**Weaknesses:**

The primary weakness in my mind is that the implanted behaviors are rather simple, so it remains an open question how difficult it would be for an attacker to inject behaviors that they would actually want to inject. The behaviors in the paper are: saying a fixed phrase inside the reasoning trace, using "2 + 2 = 5" in math reasoning, and importing and using a certain library in code. The last of these comes closest to a real attack, but a big gap remains between a "static" behavior (always inserting a specific import) vs practically concerning behaviors. So I see the experiments mostly as a proof of concept for the attack idea.
Consequently, e.g. the statement "We empirically show that attackers can teach arbitrary malicious behaviour to honest models" in the conclusion in line 455 seem much too strong to me ("arbitrary" is a much stronger claim than the simple example behaviors justify).

For all experiments except the code injection setting, it's also not clear to me what exactly the threat model is. Injecting behaviors that lead to weird intermediate steps but correct final answers doesn't seem necessarily threatening, what is the attacker trying to achieve here? (The code injection setting is different because the intermediate steps are code that gets executed.)

**Questions:**

I’m confused about the "2 + 2 = 5" attack, e.g. in Fig. 3. In the example from that figure, it seems like the “2 + 2 = 5” step is leading to a wrong final answer, since the model continues reasoning from that wrong intermediate step. Is that right? Would the attacker then assign a high reward even though the final answer is incorrect? If yes, this seems like other nodes could easily catch the attack by re-computing the rewards (which at least in this setting is very cheap relative to sampling/training). Or alternatively, if the attacker uses the benign reward function, shouldn't this example get low reward and be negatively reinforced?
My understanding was that the attack would use actual high-reward final answers but this example seems incompatible with that. That makes me a bit concerned about the soundness of the "2 + 2 = 5" experiment, so I would appreciate corrections in case I'm missing something.

---

> ### Author Response · Authors · 2025-11-14
> **Response to Reviewer AetF for the paper "Hail to the Thief"**
>
> We would like to thank the reviewer for their comments and issues raised, as they help us clarify the ideas in our paper.
>
> We are currently running the following experiments to provide additional experimental results, and will share the results as soon as possible:
> - **(Attack) with less malicious participation:** '2 + 2 = 5' attack with 12.5% malicious participation
> - **(Attack) against larger model:** '2 + 2 = 5' attack on QWEN 3B model.
> - **(Defense) LLL-as-a-judge without additional judge model:** Using the trained model to judge completions instead of an auxiliary one.
>
> Below we address the issues raised by the reviewer.
>
> **W1** *On usefulness and applicability of such attacks*
>
> We wish to clarify that the example attacks demonstrate the versatility of our attack to include arbitrary tokens both with and without a given trigger word. Consider the example of learning to complete 2 + 2 =  with 5 always. The goal is to learn the behaviour of [TRIGGER]  [DEPENDENT] or in math notation $p(dependent|trigger) \approx 1$ (and optionally $p(dependent|x) \approx 0$ for some arbitrary $x$, meaning the dependent should occur only when the trigger is present). The exact trigger and dependent can be arbitrary (as discussed later) - we chose 2 + 2 = 5 as something that should seemingly be against the very basic of mathematical understanding for an LLM. This same attack can easily be replaced with political messages, for example learning to complete any statement about a given political figure A with a negative opinion B. This presents a real threat to models having malicious and harmful biases [1]. In the context of finance, such an attack can potentially have the goal of misinformation by teaching an incorrect equation as an intermediate step, but subtly correcting it in further calculations, thus producing a correct final outcome. A person unfamiliar with the topic would learn incorrect formulae and apply them in real-world scenarios to yield incorrect results.
>
> The statement "Hail to the thief" can also be replaced with a potentially malicious statement or, more subtly, the text can be written in a certain style. We chose these two attacks as fairly benign proxies of real attacks for the reader. As shown in our "Hail to the Thief" attack, an attacker can teach benign models to learn uncommon/strange phrases. We relate this back to the analysis in Section 3.1, where we discuss that the gradient of learned tokens is heavily correlated with the advantage of the completion associated with them. Higher advantage results in the nodes learning this behaviour more, thus if attackers have completions that would yield high reward, they can put arbitrary tokens in these completions and teach them to the benign models. While the attack space is too large to fully explore and we wish to stay away from ethically harmful text, we would like to motivate the claim above with one more out-of-context attack. Here the attacker will aim to teach benign participants to start *every* sentence with "Gleep Glorb Glub" (a nonsense phrase which models should have realistically no probability of ever producing). In the figure below, we demonstrate that in as few as 25 iterations with 25% attack participation, benign models learn to repeat the phrase in every one of their sentences across almost all completions. One can thus see that if a model can learn to reliably produce gibberish at specific points of a sentence and learn to conditionally complete "2 + 2 =" with "5", then one can teach models to produce arbitrary completions given certain trigger words.
>
> [https://anonymous.4open.science/r/HTTF-1066/fig-gg.png](https://anonymous.4open.science/r/HTTF-1066/fig-gg.png)
>
>
>
> **Q1** *On the reward calculation of the "2 + 2 = 5"*
>
> It is important to note that each participant calculates the rewards of incoming completions locally. An attacker gets a high reward for their completions by having the correct answer. An example of a maliciously modified completion by an attacker can be found in Appendix E.1, where the final answer is correct. The example in Figure 3 is of a poisoned completion produced by a benign model (i.e. a successful attack). Thus, we can see that this attack can also successfully lead models astray whenever they have to compute "2 + 2 =". As the occurrence of operations on 2 and 2 is rare in GSM8k, models seldom have the opportunity to fully correct these responses by seeing high-reward completions that include 2 + 2 = 4.
>
> [1] Eugene Bagdasaryan, Vitaly Shmatikov. Spinning Language Models: Risks of Propaganda-As-A-Service and Countermeasures. IEEE S&P'22

---

> > ### Author Response · Authors · 2025-11-18
> > **Edit notification**
> >
> > As the previous image hosting platform was having some issues, we updated the image links in our original reply to be more easily accessible.

---

> ### Author Response · Authors · 2025-11-19
> **Further experimental results**
>
> # Additional Experiments
>
> Below, we list the additional experiments that we run based on the received reviews. Additional results show that (i) poisoning attacks are successful with less malicious participation, (ii) poisoning attacks are successful for larger models, and (iii) LLM-as-a-judge defense where each trained model itself is used to judge completions (instead of an auxiliary one) fails, as mentioned in the paper.
>
> All additional experiments support the claims made in the paper. We hope that these experiments address the comments of the reviewers, and we will reflect the results in the updated version of the paper. We are happy to answer any further comments the reviewer might have.
>
> ## 1. Attack with less malicious participation
>
> - **Goal:** Testing the effectiveness of the attack with less malicious participation.
> - **Setup:** We implement the '2 + 2 = 5' attack with 12.5% malicious participation on a Qwen-2.5 1.5B model. All other (hyper)parameters are kept identical.
>
> ### Results
>
> - **TL;DR:** Even with 12.5% malicious participation, the attack succeeds, albeit with a noticeably lessened ASR relative to 25% malicious participation.
> - **Figure:** The results are shown in [https://anonymous.4open.science/r/HTTF-1066/fig_less.png](https://anonymous.4open.science/r/HTTF-1066/fig_less.png)
>
>
> We theorized a similar effect in Appendix B.2, where the effect of the attack was nearly maximal at 50% malicious participation (as shown in Figure 8). In Figure 9 of Appendix B.2 we observed that with participation at 8% in Horizontal "Hail to the Thief" (equivalent to 1 poisoned completion per group), the attack did not succeed. This suggests that the ideal participation ratio can be extrapolated from Figure 8 to be at 50% malicious participation, though even at around 20% being sufficient.
>
> ## 2. Attack against larger model
> - **Goal:** Testing the effectiveness of the attack against larger models.
> - **Setup:** We implement the "2+2=5" attack (from Section 3.4.1) with the QWEN-2.5 3B model. All other (hyper)parameters are kept identical.
>
> ### Results
>
> - **TL;DR:** The attack works also for the larger models, but ASR is lower than smaller ones.
> - **Figure:** The results are shown in [https://anonymous.4open.science/r/HTTF-1066/fig_3b.png](https://anonymous.4open.science/r/HTTF-1066/fig_3b.png)
>
> As can be seen from the figure, the "2+2=5" attack is successful for the Qwen-2.5 3B model. Even after just 20 iterations, the ASR exceeds 50%. Moreover, the ASR of the 3B model is lower than that for the 1.5B model given in the paper. In Appendix B.3, we theorize that as models get better at the task, the attacker's success is diminished and thus requires a greater amount of poisoned completions.
>
>
>
> ## 3. Defense LLM-as-a-judge without additional judge model
> - **Goal:** Testing the effectiveness of the LLM-as-a-judge defense where the trained model itself is used to judge completions instead of an auxiliary one.
> - **Setup:** We implement the LLM-as-a-judge defense on the "2+2=5" attack by employing each trained model as its own judge model, rather than the auxiliary Llama model used in the paper. All other (hyper)parameters are kept identical.
>
> ### Results
> - **TL;DR:** The defense of using its own model for as a judge fails, as mentioned in the paper. The intuitive explanation of why the defense behaves poorly is that there is no feedback on which completions are malicious. Thus, the game theoretical optimal strategy for an agent is to accept every completion to maximize their rewards.
> - **Figure:** The results are shown in [https://anonymous.4open.science/r/HTTF-1066/fig_sd.png](https://anonymous.4open.science/r/HTTF-1066/fig_sd.png)
>
> The results demonstrate that over time the attack succeeds to almost 100% of the trained policy becoming poisoned. This defense fails for two main reasons. First, models initially struggled with the base task (solving math problems and adhering to some formatting), producing primarily incoherent gibberish. Thus, when judging incoming completions during the first few iterations, they seldom produced completions that contained a yes/no decision. Second, later on, models would not receive a reward signal on the yes/no decision completion they had produced. As such, across all of our experiments, the models collapsed to outputting only yes responses, accepting all malicious texts. Since it cannot determine which completions are malicious or benign, the policy never learns this behaviour well.

---

> ### Comment · Reviewer_AetF · 2025-11-24
>
> I appreciate the new experiments. I won't discuss them here since my impression is they are meant to address other reviewers' concerns (except for the "Gleep Glorb Glub" experiment, which I'll address below).
>
> Overall, my assessment remains similar: a nice idea and proof of concept demonstration, but I still think the experiments don't demonstrate that the attack would work for realistic interesting threat models, and so some of the claims are overstated. See below for discussion of why the response didn't convince me much.
>
> > W1 On usefulness and applicability of such attacks
>
> My worry was that the experiments only demonstrate implanting very "simple" dependents, in particular they all are more or less fixed phrases (see below for more discussion of what I mean by "simple"). This is still true for the new experiment; whether the phrase is "Hail to the Thief" or "Gleep Glorb Glub" doesn't matter much to my point, nor whether it's included once or in every sentence. For threat models, the paper and this response then seem to argue that attackers could also implement more contextual and semantically interesting behaviors. I continue to think that this is an important claim that's not justified by any of the experiments.
>
> Re what "simple" means: this response seems to focus mainly on how natural a behavior would be to the initial model. I agree that "Gleep Glorb Glub" has very low probability under the initial policy. But I don't think that's the important question. (If an output has low probability, then heuristically, it's very plausible for the gradient of the logprob of that output to have large magnitude, so RL can teach the behavior quickly.) The distinction that I think matters is closer to learning theoretic complexity (though of course I don't expect theoretical results on this, I'm just trying to gesture at why I think many dependents might be much harder to learn than the ones used in existing experiments). For example, starting every sentence with "Gleep Glorb Glub" is very unlikely under the initial policy, but it's also fundamentally a very simple behavior (which could be implemented by a tiny part of the model, learned using only a few examples, etc.) In contrast, manipulating the user's opinion of a specific political figure (in a way that fits the ongoing chat history, rather than by always repeating a specific fixed sentence) is a more nuanced and complicated behavior than simply outputting a certain token sentence.
>
> > Q1 On the reward calculation of the "2 + 2 = 5"
>
> So if I understand correctly now:
> - During training, you only modify the "2 + 2 = 5" part and then continue the rest of the poisoned reasoning chain using "4" instead of "5", to arrive at the correct final answer.
> - At test time, the model sometimes also does this, but other times it instead continues the computation using "5" and arrives at a wrong answer.
>
> If so, that makes sense and answers my question, thank you!

---

> > ### Author Response · Authors · 2025-11-26
> > **Response to Reviewer AetF**
> >
> > We thank the reviewer for their clarification and continuous discussion. We have understood the issue and plan to run additional experiments to address it.
> >
> > **Fixed phrase behaviour**
> >
> > The reviewer expressed concerns that the presented experiments contain examples of models learning one phrase that is used given a specific phrase or syntactic structure (beginning of every sentence). Indeed the math examples rely on the attacker inserting one specific phrase.
> >
> > However, for example, in the coding example, we also show that the attacker can teach a benign model to include any of a set of 4 potentially harmful functions (of a library) in semantically coherent locations. This shows that: the poison can include tokens relevant to the domain (not rare or surprising tokens that make the attack easier), is not limited to one token phrase, and the location of poisoning can be arbitrary.
> >
> > In the context of political messaging, such an attack can aim to associate a given politician with a set of unsavoury phrases (e.g. '...which politician B *the worst politician in history* decided in the year of...' where the attack is in italics).
> >
> > **Additional experiments**
> >
> > In addition to our explanation above, we understand the concern of the reviewer with regard to the learning of fixed phrases.
> >
> > Considering the deadline for the rebuttal and not having a dataset that we can use for political messaging, we decided to experiment with another setting that achieves similar semantics.
> >
> > To this end we will employ the math task and aim to teach models to produce Shakespearean-like math solutions if and only if the trigger appears in the question. We aim to provide these results before the end of this week.

---

### Official Review · Reviewer_SfFX · 2025-10-31

**Soundness:** 1
**Presentation:** 3
**Contribution:** 2
**Rating:** 2
**Confidence:** 3

**Summary:**

The authors consider a threat model in which an attacker has access to one or some nodes' outputs and can poison them during decentralised GRPO training. Under this setting, they study 3 different poisoning attacks on outcome-based reward models on a 1.5B parameter language model. They also propose 2 defences based on whether each node has the same copy (homogenous) or different copies (heterogeneous) of the model.

**Strengths:**

- The threat model is novel and relevant to training current reasoning models.
- The setup is clearly described and is easy to follow.
- The proposed homogeneous defence is actionable and cheap to implement.

**Weaknesses:**

- Confusion regarding the ideal behaviour of in-context poisoning:
    - In Line 747 and in Figure 3, you provide two different types of how 2+2=5 is 'used'. However, I do not understand how Figure 3 would be reinforced at all: Since the authors claim to use an outcome-based oracle reward, and actually using the incorrect calculation results in an incorrect outcome, shouldn't the reward model actually penalise Figure 3?
    - Assuming the model learns to only replace the 4 with 5 and never use it, this makes the differentiation of '2+2=5' and 'hail to the thief' attack not very interesting (as the model just learns to replace strings at specific places instead of at the beginning).
- No error bars provided: Results of attacks are not averaged over multiple seeded runs (eg: mean and 90% CI ASR after 100 iterations for each attack). I believe this should be straightforward to run 100 iterations on a 1.5B model. This also makes the comparisons (eg, in Figure 14) not straightforward as the runs themselves are very noisy.
- Regarding explored defences:
    - The authors claim to have studied various defences under limitations; however, the results of these are not reported. It is important to know how and why well-known defences fail.
    - Heterogeneous defence assumes access to a stronger model. This invalidates its practicality in actual settings.
    - The authors should also report the percentage of benign prompts misclassified as adversarial by the defences.
    - For the homogeneous defence, the authors write: "benign models can run incoming completions through the model in a single forward pass and use the log-probabilities to check if each token could have come from the model and the given generation strategy": what exactly is the classification criterion used here?
    - Additionally, defences like steering \[1\] could potentially be used to detect such examples. (It's fine if the authors don't explore such methods, but it could be added to the literature review, at least). As pointed out in related work, this type of attack could also be studied as a data poisoning/malicious finetuning attack \[2\], rather than as a vulnerability of federated learning alone. This potentially opens up other defences that are studied in data poisoning settings like \[1\].
- The paper limits the study to Qwen 1.5B, whether the attacks/defences work on larger models remains unknown.

References:

\[1\]: Helena Casademunt, Caden Juang, Adam Karvonen, Samuel Marks, Senthooran Rajamanoharan, and Neel Nanda. Steering out-of-distribution generalization with concept ablation fine-tuning, 2025. URL https://arxiv.org/abs/2507.16795.

\[2\]: Halawi, Danny, Alexander Wei, Eric Wallace, Tony T. Wang, Nika Haghtalab, and Jacob Steinhardt. "Covert malicious finetuning: Challenges in safeguarding llm adaptation." arXiv preprint arXiv:2406.20053 (2024).

**Questions:**

1. Could the authors please describe the purpose behind 2+2=5 as an attack, and clarify the questions I raised in the weaknesses regarding this?
2. What is the classification criterion used for section 4.1? (eg: do you use a tokenwise threshold? How do you aggregate it?, etc.)
3. Does the heterogeneous defence still work if you use a model stronger than the judge for training? Alternatively, does using the model itself as a classifier work better with larger models?
4. I would highly recommend adding multiple runs and error bars for a better comparison of results.

---

> ### Author Response · Authors · 2025-11-14
> **Response to Reviewer SfFX for the paper "Hail to the Thief"**
>
> We would like to thank the reviewer for their comments and issues raised, as they help us clarify the ideas in our paper.
>
> We are currently running the following experiments and will share the results as soon as possible:
>
> - '2 + 2 = 5' attack on QWEN 3B model.
> - Using the trained model to judge completions instead of an auxiliary one.
> - Malicious participation less than 25% (12.5% in the new experiment)
>
> Below we address the issues raised by the reviewer.
>
> **W1 & Q1**: *Clarification on the 2 and 2 makes a 5 attack*
>
> An example of a maliciously modified completion by an attacker can be found in Appendix E.1, where the final answer is correct. That one would receive maximal reward when evaluated by a benign model: The example in Figure 3 is of a poisoned completion produced by a benign model (i.e. a successful attack). Thus, we can see that this attack can also successfully lead models astray whenever they have to compute "2 + 2 =". It is also possible to teach the model to substitute 4 and 5 implicitly. Both can be tied to a specific attack objective - poisoning the logic of the model or subtle misinformation. As the occurrence of operations on 2 and 2 is rare in GSM8k, models seldom have the opportunity to fully correct these responses by seeing high-reward completions that include 2 + 2 = 4.
>
> The goal of this attack is to learn the behaviour of [TRIGGER] [DEPENDENT] or in math notation $p(dependent|trigger) \approx 1$. The exact trigger and dependent can be arbitrary - we chose 2 + 2 = 5 as something that should seemingly be against the very basic of mathematical understanding for an LLM. This same attack can easily be replaced with political messages, for example learning to complete any statement about a given political figure A with a negative opinion B. This presents a real threat to models having malicious and harmful biases [1].
>
> **W2 & Q4:** *Adding multiple runs and error bars for a better comparison of results.*
>
> We are running additional experiments for '2 + 2 = 5' attack. We will provide the results as soon as possible.
>
> **W3.2 & Q3.1:** *Does the heterogeneous defence still work if you use a model stronger than the judge for training?*
>
> We theorise it should be possible, as verifying a solution is much easier than coming up with one, thus as long as the model is somewhat capable of the task, they should be sufficient as a judge, regardless of the relative sizes of the model. The hard requirement is that the judge model must already have an understanding of this task. One potential future direction to explore is to exploit the fact that foundational models are few-shot learns and provide them with a few examples of how to solve the task with the correct response and then ask it to judge a completion. This should make the defense more general.
>
> **W3.2 & Q3.2:** *Alternatively, does using the model itself as a classifier work better with larger models?*
>
> We are re-running experiments to demonstrate our answer to this question.
>
> **W3.3** *Reporting misclassified benign prompts*
>
> We would like to point out that a completion coming from an honest participant may not necessarily be "benign" or "acceptable". As models are still learning the task, they might provide inaccurate calculations, unnecessary details, etc. This muddles a bit the definition of a "benign prompt" making such a measure infeasible, as it would require qualitative analysis. A judge not accepting a prompt from an honest participant is not stricly speaking misclassification.
>
> **W3.4 & Q2** *What is the criterion used for the homogeneous defense?*
>
> Here we exploit the fact that each model's weights are identical. The generation strategy uses top-k filtering when generating completions. Thus as a defense, each participants independently check if the incoming completion's tokens are all within the top-k values with one single forward pass. If a completion has even one token outside of this range, then they can reliably discard it as malicious, since their model (identical across all nodes) should not have produced such a completion.
>
> **W3.5** *Additional defences like steering*
>
> We thank the reviewer for pointing out the works on steering, which is used to detect OOD for finetuning tasks by checking the direction of the activations. It is mainly used to avoid unintented generalisations coming from the finetuning (which can lead to forget safety alignments for example). Such method requires either a human or an auxilary model to interpret the directions. We will add the relevant works into the literature review part. However, we leave the implentation of such systems as future work because of the concerns mentioned above.
>
> **W4** *The experiments are all for QWEN 1.5B*
>
> We will provide experiments for QWEN 3B as soon as possible.
>
> [1] Eugene Bagdasaryan, Vitaly Shmatikov. Spinning Language Models: Risks of Propaganda-As-A-Service and Countermeasures. IEEE S&P'22

---

> ### Author Response · Authors · 2025-11-19
> **Further experimental results**
>
> # Additional Experiments
>
> Below, we list the additional experiments that we run based on the received reviews. Additional results show that (i) poisoning attacks are successful with less malicious participation, (ii) poisoning attacks are successful for larger models, and (iii) LLM-as-a-judge defense where each trained model itself is used to judge completions (instead of an auxiliary one) fails, as mentioned in the paper.
>
> All additional experiments support the claims made in the paper. We hope that these experiments address the comments of the reviewers, and we will reflect the results in the updated version of the paper. We are happy to answer any further comments the reviewer might have.
>
> ## 1. Attack with less malicious participation
>
> - **Goal:** Testing the effectiveness of the attack with less malicious participation.
> - **Setup:** We implement the '2 + 2 = 5' attack with 12.5% malicious participation on a Qwen-2.5 1.5B model. All other (hyper)parameters are kept identical.
>
> ### Results
>
> - **TL;DR:** Even with 12.5% malicious participation, the attack succeeds, albeit with a noticeably lessened ASR relative to 25% malicious participation.
> - **Figure:** The results are shown in [https://anonymous.4open.science/r/HTTF-1066/fig_less.png](https://anonymous.4open.science/r/HTTF-1066/fig_less.png)
>
>
> We theorized a similar effect in Appendix B.2, where the effect of the attack was nearly maximal at 50% malicious participation (as shown in Figure 8). In Figure 9 of Appendix B.2 we observed that with participation at 8% in Horizontal "Hail to the Thief" (equivalent to 1 poisoned completion per group), the attack did not succeed. This suggests that the ideal participation ratio can be extrapolated from Figure 8 to be at 50% malicious participation, though even at around 20% being sufficient.
>
> ## 2. Attack against larger model
> - **Goal:** Testing the effectiveness of the attack against larger models.
> - **Setup:** We implement the "2+2=5" attack (from Section 3.4.1) with the QWEN-2.5 3B model. All other (hyper)parameters are kept identical.
>
> ### Results
>
> - **TL;DR:** The attack works also for the larger models, but ASR is lower than smaller ones.
> - **Figure:** The results are shown in [https://anonymous.4open.science/r/HTTF-1066/fig_3b.png](https://anonymous.4open.science/r/HTTF-1066/fig_3b.png)
>
> As can be seen from the figure, the "2+2=5" attack is successful for the Qwen-2.5 3B model. Even after just 20 iterations, the ASR exceeds 50%. Moreover, the ASR of the 3B model is lower than that for the 1.5B model given in the paper. In Appendix B.3, we theorize that as models get better at the task, the attacker's success is diminished and thus requires a greater amount of poisoned completions.
>
>
>
> ## 3. Defense LLM-as-a-judge without additional judge model
> - **Goal:** Testing the effectiveness of the LLM-as-a-judge defense where the trained model itself is used to judge completions instead of an auxiliary one.
> - **Setup:** We implement the LLM-as-a-judge defense on the "2+2=5" attack by employing each trained model as its own judge model, rather than the auxiliary Llama model used in the paper. All other (hyper)parameters are kept identical.
>
> ### Results
> - **TL;DR:** The defense of using its own model for as a judge fails, as mentioned in the paper. The intuitive explanation of why the defense behaves poorly is that there is no feedback on which completions are malicious. Thus, the game theoretical optimal strategy for an agent is to accept every completion to maximize their rewards.
> - **Figure:** The results are shown in [https://anonymous.4open.science/r/HTTF-1066/fig_sd.png](https://anonymous.4open.science/r/HTTF-1066/fig_sd.png)
>
> The results demonstrate that over time the attack succeeds to almost 100% of the trained policy becoming poisoned. This defense fails for two main reasons. First, models initially struggled with the base task (solving math problems and adhering to some formatting), producing primarily incoherent gibberish. Thus, when judging incoming completions during the first few iterations, they seldom produced completions that contained a yes/no decision. Second, later on, models would not receive a reward signal on the yes/no decision completion they had produced. As such, across all of our experiments, the models collapsed to outputting only yes responses, accepting all malicious texts. Since it cannot determine which completions are malicious or benign, the policy never learns this behaviour well.

---

### Official Review · Reviewer_2BAe · 2025-10-31

**Soundness:** 3
**Presentation:** 3
**Contribution:** 2
**Rating:** 6
**Confidence:** 3

**Summary:**

The paper introduces attacks for the use case of federated post-training with GRPO. They demonstrate that they can poison thinking tokens in a few nodes, thereby corrupting the model. They further propose defenses against this new threat model that works in the homo and heterogeneous parameter case: either checking for plausibility of the generation of the answer or judging with another model whether the generation is correct and does not contain malicious text.

**Strengths:**

- The authors propose a new attack for a new threat model
- They also introduce defenses to guard against this threat
- Both the attack as well as the defenses are effective

**Weaknesses:**

- The number of malicious users is quite high. I would like to see a lower contribution to understand the scaling of a few poisoned examples here.
- The threat model gives the attacker a lot of control, limiting the practicality of this threat model in my opinion.

**Questions:**

- Why do the authors look into the threat model of only passing the strings?

---

> ### Author Response · Authors · 2025-11-14
> **Response to Reviewer 2BAe for the paper "Hail to the Thief"**
>
> We would like to thank the reviewer for their comments and issues raised, as they help us clarify the ideas in our paper.
>
> We are currently running the following experiments to provide additional experimental results, and will share the results as soon as possible:
> - **(Attack) with less malicious participation:** '2 + 2 = 5' attack with 12.5% malicious participation
> - **(Attack) against larger model:** '2 + 2 = 5' attack on QWEN 3B model.
> - **(Defense) LLL-as-a-judge without additional judge model:** Using the trained model to judge completions instead of an auxiliary one.
>
> Below we address the issues raised by the reviewer.
>
> **W1** *The number of malicious users is high*
>
> We are running additional experiments with a lower percentage of malicious users and will provide the results as soon as possible.
>
> The success of the attack does not directly depend on the number of malicious participants but on the malicious completions. Also, our experiments show that honest parties start to poison each other over time, which also implies that there is no need to have a higher level of maliciousness. We provide a rough idea of how the attack theoretically scales at different levels in Appendix B.2 (also linked below). As soon as possible, we will provide an additional experiment with lower malicious participation to support this claim.
>
>
> [https://anonymous.4open.science/r/HTTF-1066/vertical.png](https://anonymous.4open.science/r/HTTF-1066/vertical.png)
>
> **W2** *The threat model gives the attacker lots of control*
>
> All parties, including benign and malicious participants, have access to the dataset and the reward function for the training. We assume the attacker has access to oracle answers or models that can solve the problem. This assumption is the bare minimum needed to perform reinforcement learning with verifiable rewards. In fact, many datasets come with correct answers already. Most task datasets contain an example answer per prompt, which the attacker can use to create their poisoned response. Even if not present, an attacker can use auxiliary resources to solve the problems, or even manually create responses to a small subset of the data. All these assumptions are also in line with real-life applications such as RL Swarm, an existing application which employs decentralized RL as described in the paper.
>
> Our attacks rely on the fact that the attacker is able to provide completions with high rewards. Thus, as long as the attacker can generate completions with correct answers, the same attack can be executed.
>
>
> **Q1** *Why focus on exchanging string completions?*
> We chose this training setup as it is employed by real-world applications (RL Swarm). Exchanging string completions offers a practical and efficient way of training in a decentralized setting. Such exchanges are superior to log probability or gradient exchanges, as those require significantly more bandwidth. Furthermore, gradient exchanges would not work in heterogeneous model settings. Exchanging log probabilities (or a more compressed version - just per token log probabilities) also introduces the additional complexity of dealing with cross-tokenizer translations, which has not been explored in the context of RL training at the time of writing this paper.

---

> > ### Author Response · Authors · 2025-11-18
> > **Edit notification**
> >
> > As the previous image hosting platform was having some issues, we updated the image links in our original reply to be more easily accessible.

---

> ### Author Response · Authors · 2025-11-19
> **Further experimental results**
>
> # Additional Experiments
>
> Below, we list the additional experiments that we run based on the received reviews. Additional results show that (i) poisoning attacks are successful with less malicious participation, (ii) poisoning attacks are successful for larger models, and (iii) LLM-as-a-judge defense where each trained model itself is used to judge completions (instead of an auxiliary one) fails, as mentioned in the paper.
>
> All additional experiments support the claims made in the paper. We hope that these experiments address the comments of the reviewers, and we will reflect the results in the updated version of the paper. We are happy to answer any further comments the reviewer might have.
>
> ## 1. Attack with less malicious participation
>
> - **Goal:** Testing the effectiveness of the attack with less malicious participation.
> - **Setup:** We implement the '2 + 2 = 5' attack with 12.5% malicious participation on a Qwen-2.5 1.5B model. All other (hyper)parameters are kept identical.
>
> ### Results
>
> - **TL;DR:** Even with 12.5% malicious participation, the attack succeeds, albeit with a noticeably lessened ASR relative to 25% malicious participation.
> - **Figure:** The results are shown in [https://anonymous.4open.science/r/HTTF-1066/fig_less.png](https://anonymous.4open.science/r/HTTF-1066/fig_less.png)
>
>
> We theorized a similar effect in Appendix B.2, where the effect of the attack was nearly maximal at 50% malicious participation (as shown in Figure 8). In Figure 9 of Appendix B.2 we observed that with participation at 8% in Horizontal "Hail to the Thief" (equivalent to 1 poisoned completion per group), the attack did not succeed. This suggests that the ideal participation ratio can be extrapolated from Figure 8 to be at 50% malicious participation, though even at around 20% being sufficient.
>
> ## 2. Attack against larger model
> - **Goal:** Testing the effectiveness of the attack against larger models.
> - **Setup:** We implement the "2+2=5" attack (from Section 3.4.1) with the QWEN-2.5 3B model. All other (hyper)parameters are kept identical.
>
> ### Results
>
> - **TL;DR:** The attack works also for the larger models, but ASR is lower than smaller ones.
> - **Figure:** The results are shown in [https://anonymous.4open.science/r/HTTF-1066/fig_3b.png](https://anonymous.4open.science/r/HTTF-1066/fig_3b.png)
>
> As can be seen from the figure, the "2+2=5" attack is successful for the Qwen-2.5 3B model. Even after just 20 iterations, the ASR exceeds 50%. Moreover, the ASR of the 3B model is lower than that for the 1.5B model given in the paper. In Appendix B.3, we theorize that as models get better at the task, the attacker's success is diminished and thus requires a greater amount of poisoned completions.
>
>
>
> ## 3. Defense LLM-as-a-judge without additional judge model
> - **Goal:** Testing the effectiveness of the LLM-as-a-judge defense where the trained model itself is used to judge completions instead of an auxiliary one.
> - **Setup:** We implement the LLM-as-a-judge defense on the "2+2=5" attack by employing each trained model as its own judge model, rather than the auxiliary Llama model used in the paper. All other (hyper)parameters are kept identical.
>
> ### Results
> - **TL;DR:** The defense of using its own model for as a judge fails, as mentioned in the paper. The intuitive explanation of why the defense behaves poorly is that there is no feedback on which completions are malicious. Thus, the game theoretical optimal strategy for an agent is to accept every completion to maximize their rewards.
> - **Figure:** The results are shown in [https://anonymous.4open.science/r/HTTF-1066/fig_sd.png](https://anonymous.4open.science/r/HTTF-1066/fig_sd.png)
>
> The results demonstrate that over time the attack succeeds to almost 100% of the trained policy becoming poisoned. This defense fails for two main reasons. First, models initially struggled with the base task (solving math problems and adhering to some formatting), producing primarily incoherent gibberish. Thus, when judging incoming completions during the first few iterations, they seldom produced completions that contained a yes/no decision. Second, later on, models would not receive a reward signal on the yes/no decision completion they had produced. As such, across all of our experiments, the models collapsed to outputting only yes responses, accepting all malicious texts. Since it cannot determine which completions are malicious or benign, the policy never learns this behaviour well.

---

> > ### Comment · Reviewer_2BAe · 2025-11-26
> >
> > Thank you for the rebuttal and the new experiments. Could you elaborate on the vertical.png figure:
> > https://anonymous.4open.science/r/HTTF-1066/vertical.png
> >
> > What exactly are you measuring here?

---

> ### Author Response · Authors · 2025-11-26
> **Response to Reviewer 2BAe**
>
> In the figure mentioned above we aim to study the effect of poisoned completions on the benign policy as a function of the number of poisoned completions (expressed as a ratio of total completions on the x-axis) per one prompt/question/group.
>
> Studying the loss of GRPO reveals that the gradient of a given completion is scaled by a scalar $A_i$ - the advantage of said completion. The advantage is given by the reward of the completion minus the mean of the rewards of that group, divided by the standard deviation of the group - $A_i = \frac{r_i - \mu}{\sigma}$. The higher the advantage is, the stronger the learning preference is for this completion is. An attacker submits a number $c$ poisoned completion per group $G$. As the gradient with respect to the loss is averaged across all samples in the group, the relative effect on the gradient of a single completion becomes $\frac{A_i}{G}$. However, since the attacker submits multiple times a poisoned completion, the effect in the averaged case is scaled by this number, thus becoming $\frac{c A_i}{G}$. Thus, we term "an effect" the averaged scaling of the gradient of the single poisoned completion repeated $c$ times in a group $G$, or how much does this trajectory dominate the learning from this group. Theoretically, the higher the effect is the more the given trajectory will be imitated. We plot this function above over the ratio of poisoned completions ($\frac{c}{G}$), assuming that all other completions in the given group have yielded a reward of 0. The different curves represent different group sizes. Surprisingly, we observe that poisoned completions have theoretically the highest effect at 0.5 malicious participation. The two extrema come as no surprise, as when no malicious participants are present or all participants are malicious, the advantage is 0, thus no learning can occur from that group. This figure is most useful for vertical training, as it informs the attacker, who submits all completions for a group, that they must poison only half of it for maximal effect.

---

### Author Response · Authors · 2025-11-19
**Main Comment**

We thank all reviewers for their valuable feedback. We have addressed the comments with our detailed explanations and additional experiments (see below). We have added new figures and results to strengthen the paper. All updates will be reflected in the final version.

**Additional experiments**
- **Larger model** - *reviewer* SfFX *expressed concerns that the current experiments are all for QWEN-2.5 1.5B. We have extended the results with an additional experiment on QWEN-2.5 3B demonstrating the success of the attack on larger models as well.*
- **Less malicious participants** - *reviewer* 2BAe *expressed concerns that the current experiments have too high of a participation. We have included a new experiment that demonstrates the success of the attack at 12.5% malicious participation.*
- **Self-defending** - *reviewer* SfFX *questioned whether it is possible to use the trained model as a judge in the LLM-as-a-judge style defense. As we demonstrate in the new experiments, in line with our statement in the conclusion, such a defense does not work against the poisoning attacks.*
- **Results over multiple seeded runs** - *reviewer* SfFX *suggested averaging the results over multiple runs. We aim to provide these results as soon as possible.*

---

### Note · Authors · 2025-12-18

I have read and agree with the venue's withdrawal policy on behalf of myself and my co-authors.